# Transcriptomic Analysis of the Reduction in Seed Oil Content through Increased Nitrogen Application Rate in Rapeseed (*Brassica napus* L.)

**DOI:** 10.3390/ijms242216220

**Published:** 2023-11-12

**Authors:** Pengfei Hao, Yun Ren, Baogang Lin, Kaige Yi, Lan Huang, Xi Li, Lixi Jiang, Shuijin Hua

**Affiliations:** 1Institute of Crop and Nuclear Technology Utilization, Zhejiang Academy of Agricultural Sciences, Hangzhou 310021, China; 11816004@zju.edu.cn (P.H.); kaige_y0113@163.com (K.Y.); huang975885829@163.com (L.H.); 202200101021@stu.xza.edu.cn (X.L.); 2Institute of Crop, Huzhou Academy of Agricultural Sciences, Huzhou 313002, China; yunhuaren@163.com; 3College of Agriculture and Biotechnology, Zhejiang University, Hangzhou 310058, China; jianglx@zju.edu.cn

**Keywords:** fatty acid, fertilizer, genetic, gene expression, protein, seed development, transcriptome

## Abstract

Nitrogen is essential for improving the seed oil yield of rapeseed (*Brassica napus* L.). However, the molecular mechanism by which increased nitrogen rates impact seed oil content is largely unknown. Therefore, a field experiment was conducted to determine how three nitrogen application rates (120, 240, and 360 kg ha^−1^) regulated seed oil content via transcriptomic analysis. The results showed that the seed yield and the protein and total N contents increased from N1 to N3, with average increases of 57.2%, 16.9%, and 79.5%, respectively. However, the seed oil content significantly decreased from N1 to N3, with an average decrease of 8.6%. These results were repeated over a number of years. The quantity of oil protein bodies observed under a transmission electron microscope was in accordance with the ultimate seed oil and protein contents. As the nitrogen application rate increased, a substantial number of genes involved in the photosynthesis, glycolysis, and phenylpropanoid biosynthesis pathways were up-regulated, as were TF families, such as AP2/ERF, MYB, and NAC. The newly identified genes were mainly involved in carbohydrate, lipid, and amino acid metabolism. Metabolic flux analysis showed that most of the genes involved in glycolysis and fatty acid biosynthesis had higher transcript levels in the early development stages. Our results provide new insights into the molecular regulation of rapeseed seed oil content through increased nitrogen application rates.

## 1. Introduction

Nitrogen is an essential element for crop growth and development and ultimately affects crop yield and quality [1,2,3]. Therefore, the modulation of nitrogen application, including the timing, amount, and fertilizer form, becomes a simple but particularly important agronomic practice in crop production. However, this becomes complex when nitrogen, as a source of fertilizer, is applied in an open-field environment. This is because interactions between nitrogen and other factors, such as water availability and soil temperature suitability, are likely to exist during crop production. Furthermore, the effects of interactions between N fertilizer and other factors on crop growth and development are much more profound than those of a single factor [4,5]. Therefore, growers must comprehensively consider various factors of nitrogen application when making a decision on the strategy for nitrogen fertilizer application.

Nitrogen can affect rapeseed growth and development. Rapeseed is an important oil crop worldwide and is mainly distributed in Canada, China, Europe, India, and Austria. China’s rapeseed-growing area is about 7.26 million hectares. However, due to the large population in China, the significant rate of vegetative oil consumption results in the substantial import of rapeseed from other countries—up to about 30% [6]. Therefore, increasing both the seed yield and oil content remains the main purpose of rapeseed breeding programs and production. Seed protein yield and fat yield are two key components of seed yield, calculated using the seed protein and fat contents. Rapeseed seed yield and oil content are important and complicated traits that are affected by both genetic control and agronomic practices, including nitrogen application. There have been large investigations into the effect of nitrogen application on rapeseed yield and oil content. The enhancement of rapeseed seed yield via nitrogen fertilizer application is a common finding, especially under nitrogen deficiency conditions. However, excessive nitrogen application is not profitable when considering comparisons between the input of nitrogen fertilizer and the increments in yield [7,8,9,10,11]. Furthermore, the risks of lodging, late maturation, and environmental pollution are high due to excessive nitrogen application [12,13]. Compared to investigations into the effect of nitrogen application on rapeseed yield, there are relatively fewer reports on the impacts of nitrogen fertilizer on seed oil content. For example, some fatty acid components, such as oleic acid, linoleic acid, and linolenic acid, were found to be significantly decreased as the nitrogen application rate increased. Furthermore, 1212 differential lipids were identified in two rapeseed varieties under different nitrogen levels [14]. Similar observations were reported on the effect of spring nitrogen application on winter rapeseed seed oil content [15,16]. Luckily, the reduction in seed oil content can be alleviated through the application of urease inhibitors, such as N-(n-butyl) thiophosphoric triamide, under different soil conditions [17]. However, Ibrahim et al. (1989) reported that the seed oil content and percentage of major fatty acid components were not significantly affected by the nitrogen rates or times [18]. Crous et al. (2021) revealed that the nitrogen fertilizer source did not affect the seed oil content, which was determined by the presence of precipitation during the flowering and seed filling stages [19]. The above results suggest that the reduction in rapeseed seed oil content caused by nitrogen application was generally consistent across the different investigations. However, some disagreements between the results might be due to the interactions between nitrogen and other factors, such as the level of soil drought. Regardless of the clear results, very little attention has been paid to the molecular regulation of the reduction in seed oil content through increased nitrogen application.

Rapeseed seed oil is a qualitative trait, and no large-effect QTLs have been found to date [20,21,22,23]. Furthermore, because seed oil biosynthesis involves a very long metabolic flow from carbohydrate glycolysis, any step during this process will lead to a low efficiency of seed oil biosynthesis. The process mainly includes photosynthetic products, which are precursors produced during carbohydrate glycolysis, such as acetyl groups for fatty acid biosynthesis in the cytosol and plastid [24]. Fatty acids are further processed, for example, through folding on the endoplasmic reticulum membrane [25]. As a result, lipids are involved in the formation of oil bodies in rapeseed seed cells. In addition to oil, protein is another important reserve in rapeseed seed cells. The relationship between the seed oil and protein content is negative [26,27]. Consequently, reductions in seed oil content are beneficial for increasing the seed protein content. The protein content in crop tissues, including seeds, is closely associated with nitrogen supply [27,28]. As a result, with an increase in nitrogen fertilizer application, the seed oil accumulation is reduced, and the protein content is enhanced. Although this phenomenon is clear and simple, how nitrogen application rates regulate seed oil and protein contents is largely unknown at the molecular level.

In the current investigation, we hypothesized that the ultra-microstructures of seeds would be different under different nitrogen application rates and seed development stages, and that unique and novel gene regulation patterns will be identified. Therefore, we applied transcriptomic analysis to reveal the effect of different nitrogen application rates on seed oil deposition. Firstly, we assayed the seed oil and protein contents under different nitrogen application rates. Then, the seed oil and protein contents were analyzed at the anatomic level using a transmission electron microscope. Lastly, we compared the rapeseed seed metabolic differences using transcriptomic analysis. Our results provide new insights into the molecular regulation of increased nitrogen application rates in the reduction in rapeseed seed oil content.

## 2. Results

### 2.1. Effects of Nitrogen Addition on Seed Yield, Oil and Protein Contents, Seed Total N Contents, and Seed Oil and Protein Yield 

According to the results, the values were similar between 2021 and 2022. Taking the results of 2022 as an analytic example, the seed yield increased with the increase in the nitrogen application level, from 2642.1 kg ha^−1^ under N1 to 4224.1 kg ha^−1^ under N3, increasing by 57.8%, while the seed yield was 3262.8 kg ha^−1^ under N2, increasing by 21.9% (Figure 1A). However, the seed oil contents decreased with the increase in nitrogen fertilizer, from 44.6% under N1 to 43.2% under N2 and 40.3% under N3, decreasing by 3.2% and 9.6% (Figure 1B). Furthermore, there was no significant difference between N1 and N2 in both growth seasons. However, as the amount of applied nitrogen further increased, the seed oil content significantly decreased as compared to N1 and N2 in both years. This result indicates that the decrease in seed oil content was not less sensitive to the seed yield. In contrast, the seed protein contents increased from 24.4% under N1 to 26.2% under N2 and 29.4% under N3 (Figure 1C), as did the total nitrogen contents, which increased from 3.3% under N1 to 4.2% under N2 and 6.1% under N3 (Figure 1D).

### 2.2. Effects of Nitrogen Addition Rates on Seed Oil Body and Protein Body Development at Different Sampling Times

In general, the changes in the seed oil content and protein content, as revealed using a transmission electron microscope (Figure 2), were in accordance with the result of the chemical analysis (Figure 1) under different nitrogen application rates. Compared with D1, the size of the oil bodies decreased and the number of small oil bodies increased at D2 and D3; the protein bodies were gradually shaped, and they increased in number with the delay in the sampling time (Figure 2). With the increase in the nitrogen application levels, the size of the oil bodies decreased under N2 and N3 in comparison with N1, while the shape of the proteins became more obvious at the same sampling time, especially at D1. Furthermore, with the delay in the sampling time, the proportion of total oil bodies seemed to decrease, while the amount of total protein bodies increased, and the outline became more obvious; this appearance was more distinct under N2 and N3 (Figure 2).

### 2.3. Transcriptomic Analysis of Rapeseed Seeds in Three Development Stages under Three Nitrogen Levels

The results from the volcano plots of the transcriptomic data showed that 698 DEGs were down-regulated and 508 DEGs were up-regulated in N2D1 in comparison with N1D1 (Figure 3A). In the comparison of N2 and N1, with the delay in the sampling time, the number of down-regulated DEGs decreased from 350 at D2 to 33 at D3, while the numbers of up-regulated DEGs were similar (Figure 3B,C). However, with the increased application of nitrogen, the differences became larger with the delay in sampling time, which showed that, in the comparison between N2 and N3, the total number of down-regulated DEGs increased from 375 at D1 to 927 at D2 and 1726 at D3, and the total number of up-regulated DEGs increased from 291 at D1 to 1414 at D2 and 3332 at D3 (Figure 3D–F).

KEGG pathway enrichment analysis showed that the down-regulated DEGs were mainly enriched in the spliceosome and protein processing in the endoplasmic reticulum pathways in the pairwise groups of N1 vs. N2 and N2 vs. N3 at D1, while the up-regulated DEGs were mainly enriched in the ribosome and DNA replication pathways in N1 vs. N2 and the plant–pathogen interaction, glycolysis, and circadian rhythm pathways in N2 vs. N3. In the comparison between N1 and N2, in terms of D2, the down-regulated DEGs were mainly enriched in the plant hormone signal transduction pathway, while the up-regulated DEGs were mainly enriched in the ribosome biogenesis in eukaryotes. Correspondingly, in the comparison of N2 and N3, the down- and up-regulated DEGs were mainly enriched in the ribosome biogenesis in eukaryotes and phenylpropanoid biosynthesis pathways, respectively. At D3, ribosome and glutathione metabolism were the main pathways enriched in N1 vs. N2 and N2 vs. N3, respectively, while the corresponding pathways were plant hormone signal transduction and photosynthesis for the up-regulated DEGs (Figure 4).

TF family classification showed that the three most common TF families were trihelix (10), NAC (8), and AP2/ERF (7) in N1 vs. N2 at D1, while HSF (6) ranked first in N2 vs. N3. At D2, in the pairwise comparison of N1 and N2, the most common TF family was AP2/ERF (14), which was far ahead of the second most common TF family, while in the comparison of N2 and N3, the three most common TF families were AP2/ERF (30), NAC (16), and MYB (12), which also had a good lead on the others. In terms of D3, in the comparison of N1 and N2, AP2/ERF (65), MYB (37), and NAC (29) were still the three most common TF families; however, no obvious superiority among the TFs was found in the comparison of N2 and N3 (Figure 5). Gene Ontology (GO)-based functional categorization analysis of the new genes revealed that carbohydrate metabolism and lipid metabolism had the most DEGs in the metabolism item, while translation and folding, sorting, and degradation were the two most common terms among all of the GO processes (Figure 6).

### 2.4. Analysis of Relative Gene Expression on the Pathway before Fatty Acid Metabolism

To ascertain the influence of different nitrogen application rates on seed oil deposition in different seed development stages, the expression levels of genes covering glycolysis to fatty acid metabolism were detected, and the results are shown in Figure 7. A large proportion of genes showed a higher expression in the early development stage than in the later development stage. In detail, the expression pattern of sucrose synthesis was not obvious, while three out of five genes showed a higher expression at D1. For UDP-Glucosepyro Phosphosphprylase, seven *UGP2* and two *UGP3* homologous were detected, while the highest values occurred at D1 and D2; this was also the case for Ribose-5-phosphate isomerase, Phosphoribulokinase, and Ribulose-1,5-bisphosphale carboxylase. The expression of Hexokinase, Glucose-6-Phosphate dehydrogenase, Pyruvate kinas, Ribose-5-phosphate isomerase, and Phosphoenolpyruvate carboxylase was similar to that of sucrose kinase, which was not very obvious, but the majority of their homologs were found to be more highly expressed at D1 and D2 than D3. As for Phosphoglucose Isomerase, Fructokinase, Trisophosphate isomerase, Glyceraldehyde-3-phosphate dehydrogenase, Phosphoglycerate kinase, Enolase, and Pyruvate Dehydrogenase, they were distinctly more highly expressed at D1.

In contrast to the others, two homologs of Malate Dehydrogenase were identified, and both showed a higher expression at D2 than D1 and D3.

### 2.5. Identification of Fatty Acid Metabolism Based on qRT-PCR

We selected seven key genes related to fatty acid synthesis and degradation and identified their expression levels under three different nitrogen levels in three seed oil development stages using qRT-PCR. In detail, the expressions of *ACCase*, *FATA*, and *DGAT*, as the three most important key genes limiting the synthesis of fatty acids, were identified via qRT-PCR. The results showed that the expression levels were decreased with the delay in the sampling time. Furthermore, the increase in the nitrogen application rates also decreased the expression of these three genes. In terms of the four key genes related to fatty acid degradation—including *ICL*, *MLS* and *KAT*—the trends of expression were the opposite—i.e., higher expression on the late sampling dates—and they also showed a higher expression with the increase in the nitrogen application rates (Figure 8).

## 3. Discussion

Rapeseed is a major crop and is a main source of vegetable oil for the food and fuel industries [30]. High seed yield and high seed oil contents are the most important traits for rapeseed cultivation; the reasonable and optimum management of nitrogen application is regarded as the most effective way to achieve these traits [31]. However, it still remains to be illustrated how N application levels influence the deposition of seeds and seed oil contents at the molecular level. In this study, transcriptomic technics were utilized to identify differences in the gene expression patterns of seeds in three different seed development stages under three different N application levels, which provided a comprehensive insight into the molecular basis of the nitrogen-regulated mechanisms of seed development.

Nitrogen fertilizer plays a role in shoot growth, energy, and reproduction [32]. Specifically, nitrogen fertilizer plays an essential role in crop physiology, shoot growth, the energy and carbohydrate status, the phytohormone balance of the plant, and crop yield [33]. In this study, the seed yield was improved with the increase in the nitrogen level from 2642.1 kg ha^−1^ under N1 to 4139.3 kg ha^−1^ under N3 (Figure 1). Several studies have reported that high rapeseed yields require high nitrogen applications to the soil, and the highest yields were acquired at 150 kg N ha^−1^ in the UK [34], 180 kg N ha^−1^ in China [35], 213 kg N ha^−1^ in Egypt [18], and 225 kg N ha^−1^ in Iran [36]. However, in our research, the highest yield was obtained at 360 kg N ha^−1^, and a linear relationship was found between the N application levels and the yields, which could be attributed to the higher number of seeds per pod and of pods per plant [36,37]. The different optimal amounts of nitrogen application rates given in these reports might be due to the different production environments, such as soil water.

However, the seed oil contents were decreased even though the yields were increased with the increase in N application, which is in agreement with previous studies [30,38,39]. The drop in the seed oil contents that occurred with the increase in the nitrogen doses may be due to the dilution of the increased seed yield by N fertilization, with the inverse relationship being found between the oil and protein contents [40]. Li et al. (2021) found that the size and number of oil bodies were related to the seed oil contents under drought treatment [41]. In this study, we can clearly identify the opposing development trends of the oil and protein bodies with the increase in the N application rates and sampling times, and that the large oil bodies gradually divided into small oil bodies, as well as protein bodies; these trends were more obvious under higher nitrogen doses, according to the TEM observations (Figure 2). However, unlike the seed yield, a significant reduction in the seed oil content was observed under N3, indicating that the yield was more sensitive to the nitrogen supply than seed oil deposition. Seed yield can be affected by nitrogen very early and in many stages and traits. For example, resistance to abiotic and biotic stresses such as insect and disease attacks, lodging resistance, and the three yield components will affect seed yield first. However, seed oil deposition occurs after anthesis, and the most common issue during seed development is the conversion, transportation, and biosynthesis of the main seed reserves, such as carbohydrates, lipids, and proteins. Therefore, the response of seed oil deposition might be slower than that of seed yield.

To elucidate the molecular mechanisms of the deposition of oil and protein in different development stages under different N doses, transcriptome technics were utilized. Here, we found that the expression of related genes was mainly high in the early development stages—namely 30–40 days after flowering—and then decreased. According to research, the development stage (30–40 days after flowering) is the rapid accumulation stage of seed oil, which will account for 70.1% to 93.5% of mature seed oil contents, moving into a slow accumulation period up to 50 days [19,41]. In this research, genes were more active at the early sampling time, D1, and then their activity gradually decreased at D2 and D3, which may be the result of the rapid synthesis and transformation of the seed oil. At the same time, the influence of different nitrogen rates on the expression of genes in different development stages was also identified using transcriptomic technics, and the key KEGG and GO categories were illustrated, including plant–pathogen interaction, glycolysis, circadian rhythm, ribosome biogenesis in eukaryotes, and the phenylpropanoid biosynthesis pathway (Figure 4). Key genes were also validated via qRT-PCR, such as ACCase, FATA, and DGAT in the fatty acid synthesis pathway and ICL, MLS, and KAT in the fatty acid degradation pathway [42]. The results showed that key fatty acid synthesis-related genes were more highly expressed at D1, and the expression levels were decreased under N2 and N3 in comparison with N1; meanwhile, the expression of fatty acid degradation-related genes showed the opposite trend, which may suggest that the more active action of fatty acid synthesis under N1 and the more active action of fatty acid degradation might contribute to the differentially accumulated seed oil contents under different nitrogen levels.

## 4. Materials and Methods

### 4.1. Experiment Locations, Plant Materials, and Field Management

This trial was conducted during the rapeseed-growing seasons of 2020–2021 and 2021–2022 under field conditions at the Zhejiang Academy of Agricultural Sciences (30.32 N, 120.20 E), Hangzhou, China. The experiment location has a subtropical monsoon climate, with an average annual rainfall of 1150 mm-1550 mm and an average annual temperature of 15.3–16.2 °C. A traditional rapeseed (*Brassica napus* L.) variety, Zheyou50, was used as the plant material, representing a leading variety in the downstream of the Yangtze River basin. The soil type was loamy clay (hydragric), and the previous crop was rice. The soil pH, organic matter content, total N content, available P content, and available K content were 5.9, 23.5 g kg^−1^, 1.9 g kg^−1^, 135.0 mg kg^−1^, and 274.5 mg kg^−1^, respectively.

Basel fertilizers including K_2_O (54%), P_2_O_5_ (12%), and Na_2_B_4_O_7_ (99.8%) were applied manually at the rate of 120, 375, and 15 kg ha^−1^, respectively. Five to six seeds were sown into each shallow hole directly at a depth of 3 cm in each plot and then covered by a layer of soil. An amount of 750 mL of s-metolachlor (96%, *v*/*v*) with 1500 kg ha^−1^ of water was sprayed to control weeds. Other field management practices, such as pest and disease control, were performed according to local traditional practices. Excess seedlings were removed, and only one plant remained after 1 month. No irrigation was applied during the whole growing season.

### 4.2. Experimental Design

Urea was applied as a nitrogen fertilizer with three levels, which were low nitrogen (N1, 120 kg ha^−1^), optimal nitrogen (N2, 240 kg ha^−1^), and high nitrogen (N3, 360 kg ha^−1^). Here, 70% of the urea was applied to the soil as a basic nitrogen fertilizer, and 30% of the urea was applied on top in the budding stage. Rapeseed seeds under three nitrogen levels were sampled 30 days (D1), 40 days (D2), and 50 days (D3) after flowering for further analysis. Three replicates were set for each treatment, and completely randomized block methods were used. The sowing date was 1st October. The area of each block was 30 m^2^, and the density of plants was 120,000 per hectare. In detail, the size of each plot was 10 m^2^, and 1000 seeds per plot were sowed, with 250 plants per plot being kept. The same field management practices were conducted for the different treatments during the whole growing season.

### 4.3. Measurement of Seed Yield and Seed Oil Contents

In the yellow ripening stage, the siliques of rapeseed plants were harvested, and seeds were threshed out manually. The weight of the seeds after filtering was identified as the seed yield. The seed oil contents were measured using near-infrared reflectance spectroscopy (NIRS) methods. In brief, pure dry mature rapeseed seeds were filled in specific glass containers for scanning, and an ANTARIS II FT-NIR Analyzer (ThermoFisher, Waltham, MA, USA) was used to identify the oil contents according to Kaur et al. (2017) [43].

### 4.4. Measurement of Seed Protein Contents and Total Nitrogen Contents

The total nitrogen content of the seeds was determined via the Kjeldahl digestion method using an Automatic Kjeldahl Nitrogen Analyzer (ZDDN-II-B, Zhejiang topYUNNONG biotech, Hangzhou, China). Seed protein contents were measured using assay kits purchased from Sangon Biotech (Shanghai, China), and the measurement process was performed according to the manufacturer’s instructions, based on the Kjeldahl method according to Hao et al. (2021) [44].

### 4.5. Observation of the Development of Oil and Protein Bodies under a Transmission Electron Microscope

Rapeseed seeds under the three nitrogen levels were tagged and sampled 30 days, 40 days, and 50 days after flowering, and the seed cotyledons were isolated and stored in 2.5% glutaraldehyde solutions. Osmium tetroxide treatment, dehydration, epoxy resin embedding, ultrathin sectioning, and observation were conducted according to Li et al. (2021) [41].

### 4.6. RNA Extraction, and Transcriptomic Analysis

Seed samples under the 3 different nitrogen treatments and 3 different development stages were collected and stored at −80 °C, and total RNA was extracted using an RNA isolation kit (Bioteke, Beijing, China). Library construction was conducted according to the manufacturer’s instructions, sequencing was performed on an Illumina Nova-seq 6000 system, and a HISAT2 was used to align all reads against the reference genome according to Hao et al. (2022) [45]. Fragments per kilobase of transcript per million fragments mapped (FPKM) values were used to determine the expression levels of genes, and |log_2_FoldChange| > 1 and *p*-value ≤ 0.05 were used to identify differentially expressed genes (DEGs) using the edge R program and DEseq2 methods [46]. Bioinformatic analysis and the expression trends of each gene under the three N treatments and three development stages were presented in a heatmap after centralization and standardization using the R language according to Hao et al. (2022) [45].

### 4.7. qRT-PCR Identification

Primers were synthesized by Sangon Biotech Co., Ltd. qRT-PCR was performed using a CFX96^TM^ Real-time PCR Detection Instrument (BIO-RAD, Hercules, CA, USA) according to Hao et al. (2023) [47], and the expression level was calculated via 2^−ΔΔCT^. The primer sequences are shown in Appendix A.

### 4.8. Statistics

The IBM SPSS v.22.0 statistical software (SPSS, Chicago, IL, USA) was used for data analysis, and evaluation of significant treatment effects was performed using Duncan’s multiple range test (DMRT). OriginPro v.2023b was used for graph drawing. The data are described as the mean ± SD. A value of *p* < 0.05 was considered as the significance limit for all comparisons.

## 5. Conclusions

In the current study, we performed a comparative analysis of transmission electron microscope observations and transcriptome analysis based on different N levels and seed development stages, and the ultimate seed yield, seed oil yield, and seed protein yield were measured, revealing significant implications of N management in rapeseed production. Additionally, we provided a comprehensive N-regulated seed oil and seed protein development gene map, which showed the significant influence of genetic improvement on rapeseed yield under N management.

## Figures and Tables

**Figure 1 ijms-24-16220-f001:**
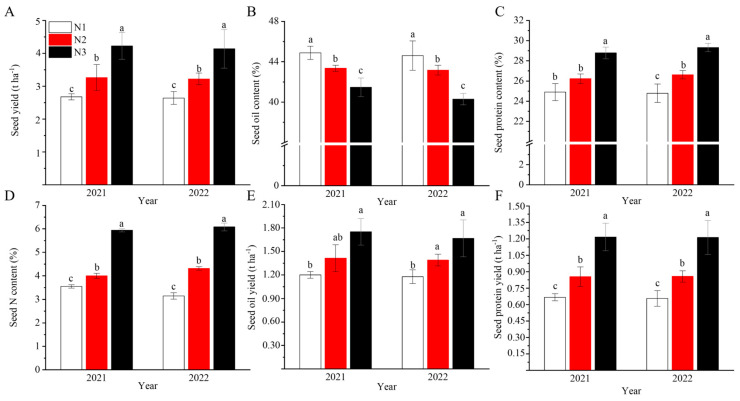
Influences of nitrogen application rates on seed yield (**A**), seed oil content (**B**), seed protein content (**C**), total N content (**D**), seed oil yield (**E**), and seed protein yield (**F**) in the 2021 and 2022 rapeseed growth seasons. N1, N2, and N3 represent three nitrogen application levels, which were 120, 240, and 360 kg ha^−1^. Different lowercase letters indicate a significant difference among the treatments using Duncan’s method (*p* < 0.05). Error bars indicate the SD values.

**Figure 2 ijms-24-16220-f002:**
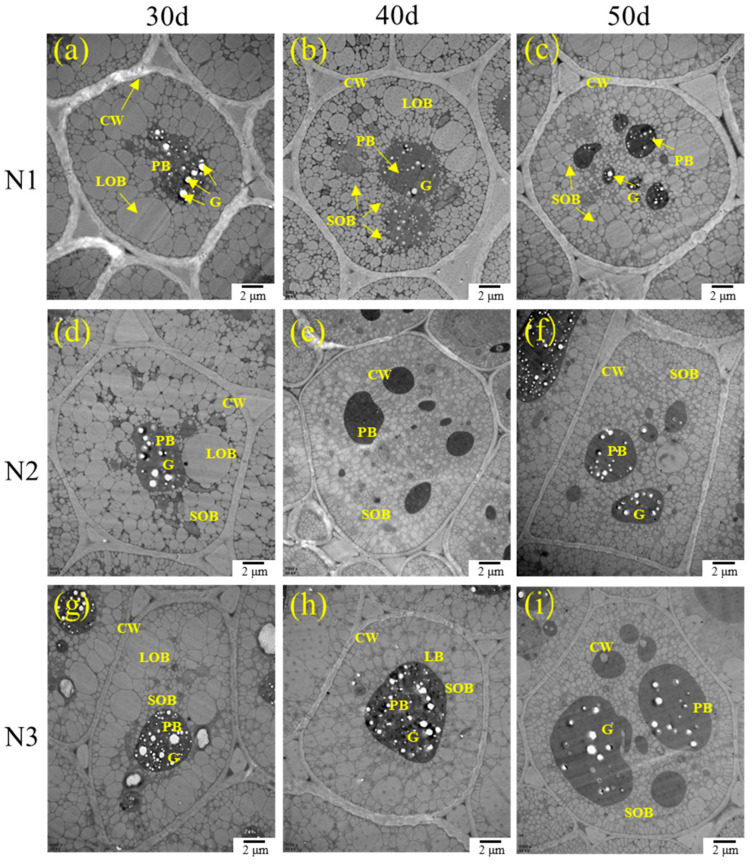
Influences of nitrogen application rates on oil bodies and protein contents in developing seeds 30, 40, and 50 days after anthesis (DAA), as observed under a transmission electron microscope. (**a**): 30DAA-N1; (**b**): 40DAA-N1; (**c**) 50DAA-N1; (**d**): 30DAA-N2; (**e**): 40DAA-N2; (**f**): 50DAA-N2; (**g**): 30DAA-N3; (**h**): 40DAA-N3; (**i**): 50DAA-N33. CW, cell wall; LOB, large oil body; SOB, small oil body; PB, protein body; G, globoid. N1, N2 and N3 represented three nitrogen application levels, which were 120, 240, and 360 kg ha^−1^.

**Figure 3 ijms-24-16220-f003:**
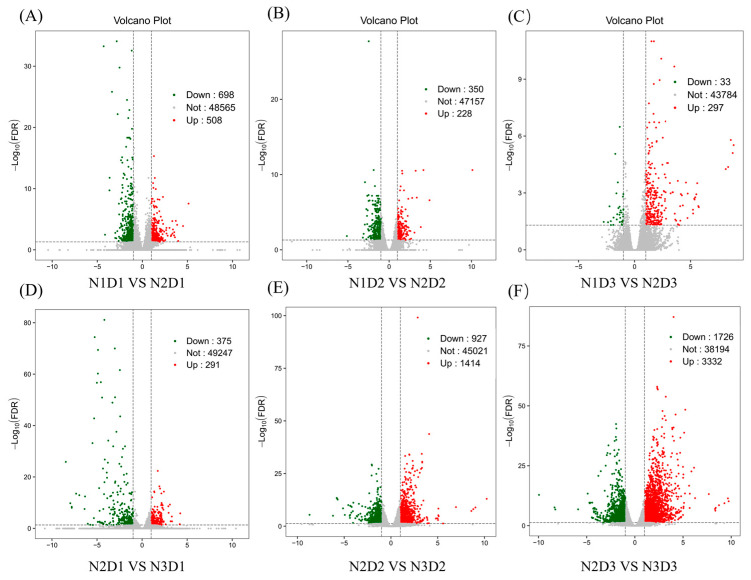
Volcano plots of differentially expressed genes at different sampling times under different N application rates. (**A**) N1D1 vs. N2D1; (**B**) N1D2 vs. N2D2; (**C**) N1D3 vs. N2D3; (**D**) N2D1 vs. N3D1; (**E**) N2D2 vs. N3D2; (**F**) N2D3 vs. N3D3. Each dot in the volcano plot represents a gene. The X axis indicates the log value of expression differential folds of a gene between two samples. The Y axis indicates the minor log value of the false discovery rate (FDR). The greater the absolute value of the X axis, the larger the expression differential folds. The greater the value of the Y axis, the more significant and reliable the differentially expressed gene. The green dots in the plot indicate down-regulated differentially expressed genes, the red dots in the plot indicate up-regulated differentially expressed genes, and the gray dots in the plot indicate non-differentially expressed genes. N1, N2, and N3 represent three nitrogen application levels, which were 120, 240, and 360 kg ha^−1^. D1, D2, and D3 represent three sampling stages, which were 30, 40, and 50 days after anthesis.

**Figure 4 ijms-24-16220-f004:**
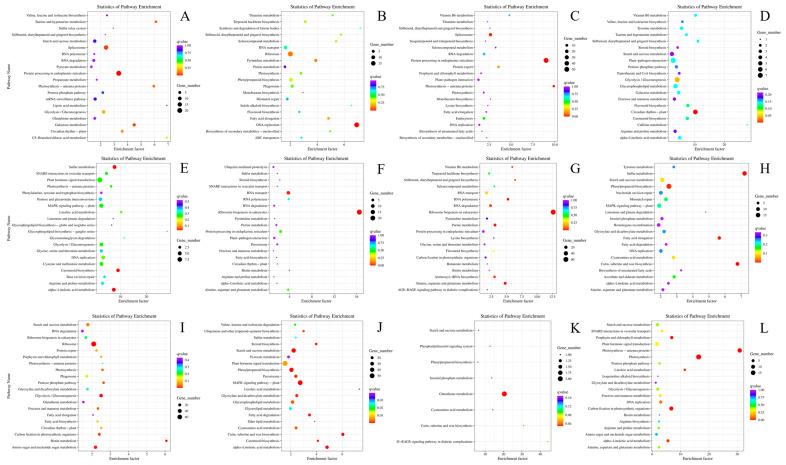
KEGG analysis of differentially expressed genes at different sampling times under different N application rates. (**A**) N1D1 vs. N2D1, down-regulated genes; (**B**) N1D1 vs. N2D1, up-regulated genes; (**C**) N2D1 vs. N3D1, down-regulated genes; (**D**) N2D1 vs. N3D1, up-regulated genes; (**E**) N1D2 vs. N2D2, down-regulated genes; (**F**) N1D2 vs. N2D2, up-regulated genes; (**G**) N2D2 vs. N3D2, down-regulated genes; (**H**) N2D2 vs. N3D2, up-regulated genes; (**I**) N1D3 vs. N2D3, down-regulated genes; (**J**) N1D3 vs. N2D3, up-regulated genes; (**K**) N2D3 vs. N3D3, down-regulated genes; (**L**) N2D3 vs. N3D3, up-regulated genes. N1, N2, and N3 represent three nitrogen application levels, which were 120, 240, and 360 kg ha^−1^. D1, D2, and D3 represent three sampling stages, which were 30, 40, and 50 days after anthesis.

**Figure 5 ijms-24-16220-f005:**
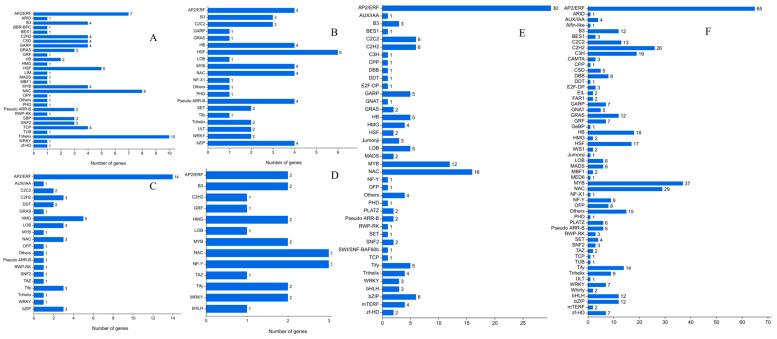
Identification of differentially expressed TFs in response to different N application rates and different development stages of seeds. (**A**) N1D1 vs. N2D1; (**B**) N1D2 vs. N2D2; (**C**) N1D3 vs. N2D3; (**D**) N2D2 vs. N3D2; (**E**) N2D1 vs. N3D1; (**F**) N2D3 vs. N3D3. N1, N2, and N3 represent three nitrogen application levels, which were 120, 240, and 360 kg ha^−1^. D1, D2, and D3 represent three sampling stages, which were 30, 40, and 50 days after anthesis.

**Figure 6 ijms-24-16220-f006:**
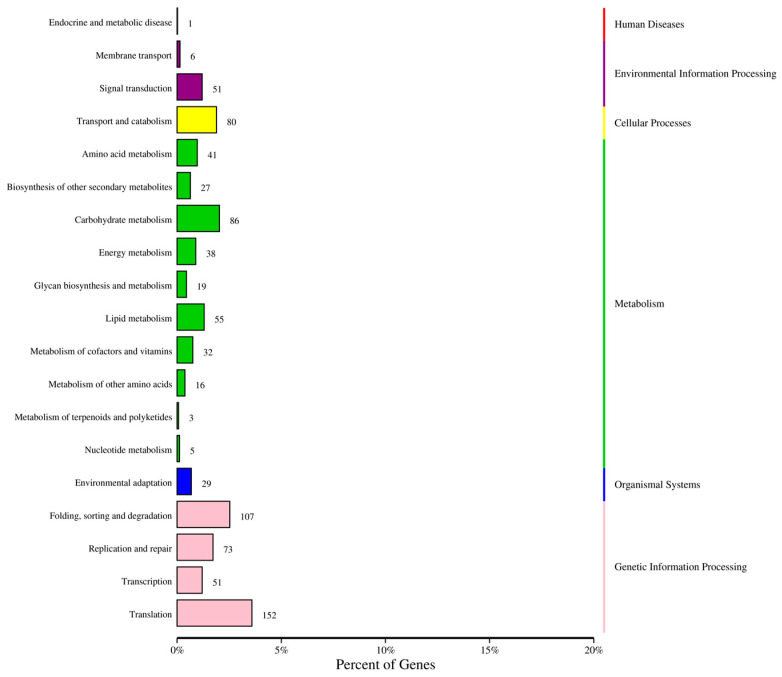
Identification of new differentially expressed genes in response to different N application rates and different development stages of seeds.

**Figure 7 ijms-24-16220-f007:**
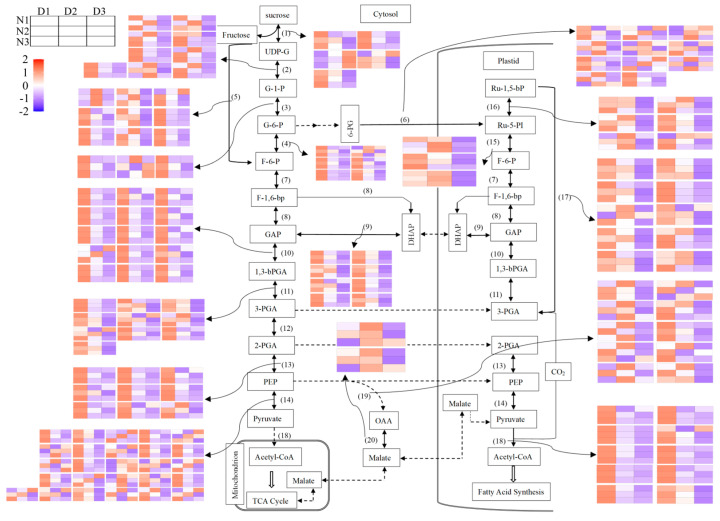
Identification of the expression of genes that were related to the pathways that come before fatty acid synthesis. The skeleton of the illustration was adopted from Houston et al. (2009) [29]. The enzymes in each step are listed as follows: (1) Sucrose Synthase; (2) UDP-Glucosepyro Phosphosphprylase; (3) Hexokinase; (4) Phosphoglucose Isomerase; (5) Fructokinase; (6) Glucose-6-Phosphate dehydrogenase; (7) 6-Phosphofructokinas; (8) Fructose 1,6 bisphosphate aldolase; (9) Trisophosphate isomerase; (10) Glyceraldehyde-3-phosphate dehydrogenase; (11) Phosphoglycerate kinase; (12) Phosphoglyceromutase; (13) Enolase; (14) Pyruvate kinase; (15) Ribose-5-phosphate isomerase; (16) Phosphoribulokinase; (17) Ribulose-1,5-bisphosphale carboxylase; (18) Pyruvate Dehydrogenase; (19) Phosphoenolpyruvate carboxylase; (20) Malate Dehydrogenase. The box in the horizontal direction, from left to right, represents the three sampling stages, which were D1, D2, and D3, while the box in the vertical direction, from top to bottom, represents the three nitrogen application levels, which were N1, N2, and N3. N1, N2, and N3 represent three nitrogen application levels, which were 120, 240, and 360 kg ha^−1^. D1, D2, and D3 represent three sampling stages, which were 30, 40, and 50 days after anthesis. The red in the box indicates up-regulated genes, while the blue in the box indicates down-regulated genes. The deeper the red or blue, the stronger the up- or down-regulation of the gene.

**Figure 8 ijms-24-16220-f008:**
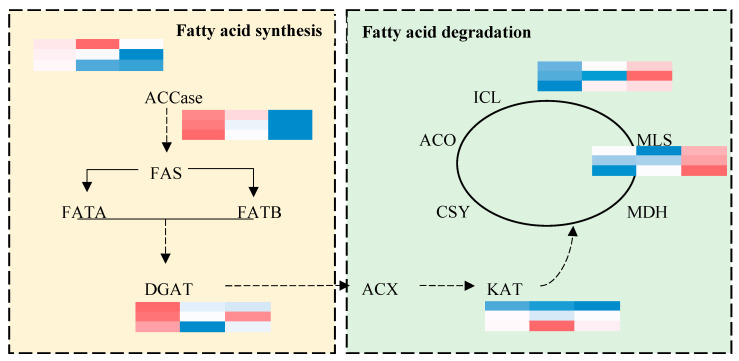
Expression levels of genes related to fatty acid synthesis and degradation identified via qRT-PCR. ACCase, Acetyl-coA carboxylase; FAS, fatty acid synthesis; FATA, Acyl-ACP thioesterase A; FATB, Acyl-ACP thioesterase B; DGAT, Diacylglycerol aeyltransferase; ACX, Acylase A oxidase; KAT, 3-Ketoacyl-coA thiolase; MDH, Malicacid dehydrogenase; MLS, Malate synthesis; ICL, Isocitrate lyase; ACO, Aconitase CSY; Citrate synthesis.

## Data Availability

The data are available upon request.

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
