# Peer review of "Transcriptomic Analysis of the Reduction in Seed Oil Content through Increased Nitrogen Application Rate in Rapeseed (Brassica napus L.)"

_ijms, 2023, doi:10.3390/ijms242216220_

Round 1
Reviewer 1 Report
Comments and Suggestions for Authors
The manuscript titled "Transcriptomic analysis on the reduction of seed oil content by increased nitrogen application rate in rapeseed (Brassica napus L.)" contains interesting research results for science and agricultural practice. I appreciate that this is a two-year field study. However, the publication requires correction and improvement. I have included detailed comments in the original PDF text. After making corrections, I recommend publishing the manuscript in the journal Int. J. Mol. Sci.
General notes:
write in the abstract whether the results obtained were repeatable over the years
add: protein to keywords
in the introduction, write about the protein yield and fat yield (i.e. by calculating the seed yield and fat content in the seeds)
in the results, you can add a new subsection regarding protein yield and fat yield (I leave it to the authors' discretion)
add reference (Table 8) in the text
I suggest adding chapter 5 Conclusions (additionally write what further research should be)
The Material and Methods chapter is the weakest (notes in the PDF text)
Complete: Supplementary Materials and Author Contributions
Correct the list of references as required by the jornal
I hope that my comments will help the authors improve the text of the manuscript. Thank you for your cooperation

Author Response
Dear reviewer
Many thanks for your comments and kind help to improve our manuscript quality. We have thoroughly revised manuscript as you suggested.
The manuscript titled "Transcriptomic analysis on the reduction of seed oil content by increased nitrogen application rate in rapeseed (Brassica napus L.)" contains interesting research results for science and agricultural practice. I appreciate that this is a two-year field study. However, the publication requires correction and improvement. I have included detailed comments in the original PDF text. After making corrections, I recommend publishing the manuscript in the journal Int. J. Mol. Sci.
Author’s answer: Thanks a lot for your appreciation to our findings and kindly proposals for the improvements of our manuscript, thank you very much. And the manuscript was revised accordingly.
General notes:
write in the abstract whether the results obtained were repeatable over the years
Author’s answer: thanks for your suggestion, the abstract was shortened, and ‘These results were repeatable over the years’ were added in the abstract.
add: protein to keywords
Author’s answer: thanks for your suggestion, ‘protein’ was added as keyword.
in the introduction, write about the protein yield and fat yield (i.e. by calculating the seed yield and fat content in the seeds)
Author’s answer: thanks for your suggestion, the description of protein and fat yield was added in Line 47-48 according to your suggestion.
in the results, you can add a new subsection regarding protein yield and fat yield (I leave it to the authors' discretion)
Author’s answer: thanks for your suggestion, the seed oil and seed protein yield per hectare were calculated by multiplying the seed yield by the oil or protein contents and divide by 100 according to your advice, and the related figures were added in Figure 1E and Figure 1F.
add reference (Table 8) in the text
Author’s answer: thanks for your suggestion, the reference was added in part 2.5.
I suggest adding chapter 5 Conclusions (additionally write what further research should be)
Author’s answer: thanks for your suggestion, the last paragraph was set as Conclusion, and the discussion about what further research should be was added.
The Material and Methods chapter is the weakest (notes in the PDF text)
Author’s answer: thank you very much for your careful review, which were useful for manuscript improvement, the soil conditions, field management and weather conditions were added in 5.1, the detail information of the experiment plot and harvest stage were added in 5.2 and 5.2 and the citation format was checked.
Complete: Supplementary Materials and Author Contributions
Author’s answer: thank you very much for your careful review, the title of Table S1 was added and the author contributions was modified.
Correct the list of references as required by the journal
Author’s answer: thanks for your suggestion, the reference format was checked.
I hope that my comments will help the authors improve the text of the manuscript. Thank you for your cooperation.
Author’s answer: thanks for your suggestion, your comments are very precious, and help us a lot on the improvement of our manuscript, we have modified our manuscript accordingly.
Reviewer 2 Report
Comments and Suggestions for Authors
Title: Transcriptomic analysis on the reduction of seed oil content by increased nitrogen application rate in rapeseed (Brassica napus L.)
Dear Authors
The MS is very interesting and fall within the scope of the journal. The experimental dataset undoubtedly are useful and constitutes scientific values.
The aim this study was applied transcriptomic analysis to reveal the effect of nitrogen application rate on seed oil deposition. Comparison of metabolic differences in rapeseed was performed using transcriptomic analysis. Manuscript is written correctly and does not raise any major objections.
Remarks
§ Subsection 2.1. - The title should be improved.
§ Figure 1A - The rapeseed seed yield should be given in t ha-1 or Mg ha-1.
§ The readability of Figures 3, 4, 5, and 7 needs to be improved.
§ Lines 323-327 should be saved as a separate Conclusions section. It needs to be expanded. Please also include prospects for further research here.
§ Subsection 4.1. - Please provide the soil type according to WRB. Line 335 - provide the percentage of ingredients in the fertilizer. Please standardize the recording of compounds in fertilizers. In subsection 4.1. physical and chemical properties of the soil should be added. This information is very important when applying urea to the soil. The use of high doses of nitrogen in research is a significant threat to the quality of rapeseed seeds.
§ The References section should be adapted to the publishing requirements. Please complete the doi numbers.
Best regards
Author Response
Dear Reviewer
Very thank you for your efforts and kind help to improve our manuscript quality. We have followed your suggestions and revised thoroughly in the text. We hope our revision will meet your standard. Thank you again.
The MS is very interesting and fall within the scope of the journal. The experimental dataset undoubtedly are useful and constitutes scientific values.
The aim this study was applied transcriptomic analysis to reveal the effect of nitrogen application rate on seed oil deposition. Comparison of metabolic differences in rapeseed was performed using transcriptomic analysis. Manuscript is written correctly and does not raise any major objections.
Author’s answer: Thanks a lot for your recognition to our work, and we are extremely grateful to your carefully review, positive comments and constructive suggestions. The manuscript was revised carefully according to your suggestions.
Remarks
- Subsection 2.1. - The title should be improved.
Author’s answer: Thanks for your comments, this title was modified as “Effects of nitrogen addition on seed yield, oil and protein contents, seed total N contents, and seed oil and protein yield.”
- Figure 1A - The rapeseed seed yield should be given in t ha-1or Mg ha-1.
Author’s answer: Thanks for your comments, the unit was changed to t ha-1.
- The readability of Figures 3, 4, 5, and 7 needs to be improved.
Author’s answer: Thanks for your comments, figures 3, 4, 5 and 7 were improved for readability. The order of TFs in Figure 5 was rearranged.
- Lines 323-327 should be saved as a separate Conclusions section. It needs to be expanded. Please also include prospects for further research here.
Author’s answer: Thanks for your comments, this part was separated as Conclusion section, and the discussion was expanded accordingly.
- Subsection 4.1. - Please provide the soil type according to WRB.
Author’s answer: Thanks for your comments, the soil type was added in part 5.1.
Line 335 - provide the percentage of ingredients in the fertilizer. Please standardize the recording of compounds in fertilizers.
Author’s answer: Thanks for your suggestion, the description of fertilizer was modified.
In subsection 4.1. physical and chemical properties of the soil should be added. This information is very important when applying urea to the soil. The use of high doses of nitrogen in research is a significant threat to the quality of rapeseed seeds.
Author’s answer: Thanks for your suggestion, the basic soil properties was added in part 5.1.
- The References section should be adapted to the publishing requirements. Please complete the doi numbers.
Author’s answer: Thanks for your suggestion, the doi numbers were added.
Best regards
Thanks again for your kindly review, and best wishes.
Reviewer 3 Report
Comments and Suggestions for Authors
The manuscript titled “Transcriptomic analysis on the reduction of seed oil content by increased nitrogen application rate in rapeseed (Brassica napus L.)" presents original research findings that significantly contribute to the fields of plant sciences and plant physiology. The primary objectives of this study were to investigate how nitrogen application rates influence seed oil and protein content. The experiments were conducted under field conditions, involving measurements of physiological and morphological features. The results of this research work expand our understanding of these aspects. Undoubtedly, this manuscript falls within the scope of the International Journal of Molecular Sciences and contains intriguing studies. It offers valuable insights and can serve as a crucial source of information. The introduction is well-crafted, and the materials and methods section adequately covers the necessary elements, detailing the experimental preparations and analyses. The data analysis is generally comprehensive (please see comments below), and the results provide valuable information. The obtained data are adequately discussed.
However, it is essential to address some shortcomings identified by the reviewers before the publication of this work.
1) Authors did not make a research hypothesis.
2) Abstract: this part should be a maximum of 200 words, but it is over 350.
3) Keywords: they cannot be this same as in title.
4) MM section, statistic analysis: Have all the criteria been met for parametric test to be used? equality of variances, normality of distribution?
5) MM section: I suggest to add GPS location data of the field where the experiment was conducted.
6) MM section: The weather conditions data should be added (such as rainfall, temperature, etc.).
7) Results: in the captions of figure 1 there is no information what the error bars mean (SD or SE?)
8) I suggest to used literature about physiology of rape plants:
Stachurska et al. Deacclimation-Induced Changes of Photosynthetic Efficiency, Brassinosteroid Homeostasis and BRI1 Expression in Winter Oilseed Rape (Brassica napus L.)—Relation to Frost Tolerance. Int. J. Mol. Sci. 2022, 23(9), 5224; https://doi.org/10.3390/ijms23095224
I would like to underline that my remarks are auxiliary and not undertake the quality and importance of the paper.
Author Response
Dear reviewer
Very thank you for your comments and kind help to improve our manuscript quality. We have revised thoroughly following your suggestions. We hope that our revision can meet your requirement. Thank you once again.
The manuscript titled “Transcriptomic analysis on the reduction of seed oil content by increased nitrogen application rate in rapeseed (Brassica napus L.)" presents original research findings that significantly contribute to the fields of plant sciences and plant physiology. The primary objectives of this study were to investigate how nitrogen application rates influence seed oil and protein content. The experiments were conducted under field conditions, involving measurements of physiological and morphological features. The results of this research work expand our understanding of these aspects. Undoubtedly, this manuscript falls within the scope of the International Journal of Molecular Sciences and contains intriguing studies. It offers valuable insights and can serve as a crucial source of information. The introduction is well-crafted, and the materials and methods section adequately covers the necessary elements, detailing the experimental preparations and analyses. The data analysis is generally comprehensive (please see comments below), and the results provide valuable information. The obtained data are adequately discussed. However, it is essential to address some shortcomings identified by the reviewers before the publication of this work.
Author’s answer: Thank you for your high recognition to our work, and great appreciation to your kindly and carefully comments on our manuscript, which were very helpful for its improvement.
- Authors did not make a research hypothesis.
Author’s answer: thanks for your comments, the research hypothesis was added in Line 93-95.
- Abstract:this part should be a maximum of 200 words, but it is over 350.
Author’s answer: thanks for your comments, the abstract was shortened.
- Keywords:they cannot be this same as in title.
Author’s answer: thank you for your suggestion, the key words were changed accordingly.
- MM section, statistic analysis:Have all the criteria been met for parametric test to be used? equality of variances, normality of distribution?
Author’s answer: thanks for your comments, the statistical analysis was improved.
- MM section:I suggest to add GPS location data of the field where the experiment was conducted.
Author’s answer: thanks for your suggestion, the detail location of the field trials was added at part 5.1.
- MM section:The weather conditions data should be added (such as rainfall, temperature, etc.).
Author’s answer: thanks for your suggestion, the description of weather conditions was added at part 5.1.
- Results:in the captions of figure 1 there is no information what the error bars mean (SD or SE?)
Author’s answer: thanks for your suggestion, the description of the error bars was added.
8) I suggest to used literature about physiology of rape plants:
Stachurska et al. Deacclimation-Induced Changes of Photosynthetic Efficiency, Brassinosteroid Homeostasis and BRI1 Expression in Winter Oilseed Rape (Brassica napus L.)—Relation to Frost Tolerance. Int. J. Mol. Sci. 2022, 23(9), 5224; https://doi.org/10.3390/ijms23095224
Author’s answer: thank you for your suggestion, this is a good study which answered the question of how the deacclimation process affects frost tolerance, photosynthetic efficiency, brassinosteroid (BR) homeostasis and BRI1 expression of winter oilseed rape. We have added the citation of this research in the Discussion.
I would like to underline that my remarks are auxiliary and not undertake the quality and importance of the paper.
Thanks again for your appreciation and kindly review to our work, and best wishes for you.